# MAmmoTH: Building Math Generalist Models through Hybrid Instruction Tuning

♣Xiang Yue*, ‡Xingwei Qu, †Ge Zhang, °Yao Fu, §Wenhao Huang,
♣Huan Sun, ♣Yu Su, †Wenhu Chen*

†University of Waterloo, ♣The Ohio State University, ‡HKUST, °University of Edinburgh, §01.AI
`yue.149@osu.edu, wenhuchen@uwaterloo.ca`

## Abstract

We introduce `MAmmoTH`, a series of open-source large language models (LLMs) specifically tailored for general math problem-solving. The `MAmmoTH` models are trained on `MathInstruct`, our meticulously curated instruction tuning dataset. `MathInstruct` is compiled from 13 math datasets with intermediate rationales, six of which have rationales newly curated by us. It presents a unique hybrid of chain-of-thought (CoT) and program-of-thought (PoT) rationales, and also ensures extensive coverage of diverse fields in math. The hybrid of CoT and PoT not only unleashes the potential of tool use but also allows different thought processes for different math problems. As a result, the `MAmmoTH` series substantially outperform existing open-source models on nine mathematical reasoning datasets across all scales with an average accuracy gain between 16% and 32%. Remarkably, our `MAmmoTH`-7B model reaches 33% on MATH (a competition-level dataset), which exceeds the best open-source 7B model (WizardMath) by 23%, and the `MAmmoTH`-34B model achieves 44% accuracy on MATH, even surpassing GPT-4's CoT result. Our work underscores the importance of diverse problem coverage and the use of hybrid rationales in developing superior math generalist models.

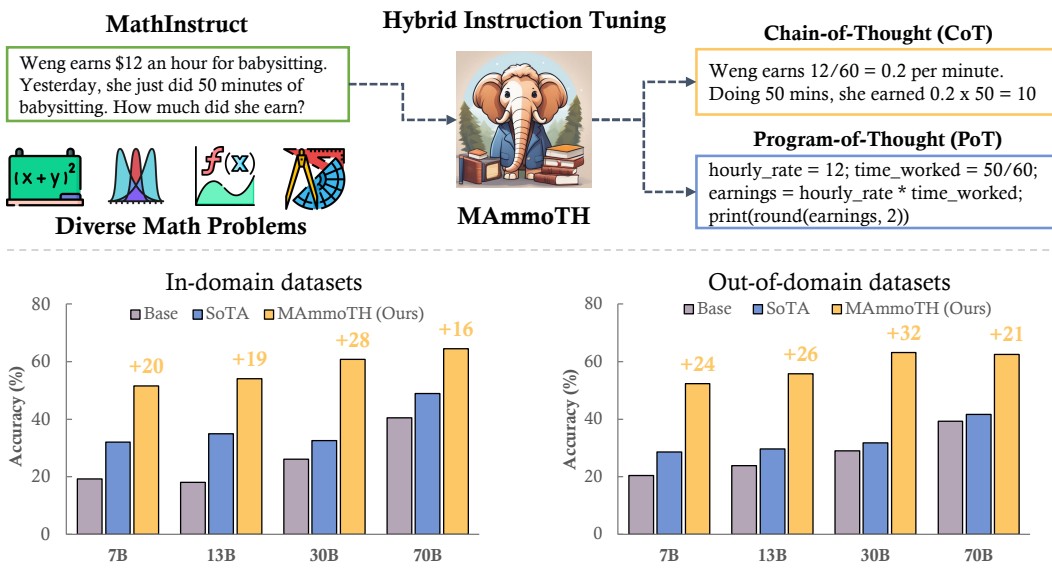

Figure 1: The superior performance of `MAmmoTH`, a series of models instruction-tuned to solve a diverse set of mathematical problems using hybrid CoT and PoT rationales. `MAmmoTH` significantly outperforms base and SoTA models on both in-domain and out-of-domain test sets, across all scales.

---

*Xiang Yue and Wenhu Chen are the leading authors of the paper. They contributed equally to this project.

# 1 INTRODUCTION

This work focuses on mathematical reasoning, a critical capability of modern large language models (LLMs) (OpenAI, 2023; Anil et al., 2023). Despite the recent advances in this field, a noticeable gap exists between closed-source and open-source LLMs—closed-source models like GPT-4 (OpenAI, 2023), PaLM-2 (Anil et al., 2023), and Claude 2 (Bai et al., 2022) dominate popular mathematical reasoning benchmarks such as GSM8K (Cobbe et al., 2021) and MATH (Hendrycks et al., 2021b), while open-source models like Llama (Touvron et al., 2023a;b), Falcon (Penedo et al., 2023), OPT (Zhang et al., 2022) lag behind on all benchmarks by a wide margin.

Current efforts to bridge this gap are twofold: (1) *Continued pre-training* like Galactica (Taylor et al., 2022) and MINERVA (Lewkowycz et al., 2022), which continues to train an LLM on math-related web data of more than 100B tokens. This approach improves a model's general scientific reasoning capability but incurs a high computation cost. (2) *Dataset-specific fine-tuning* like rejection sampling fine-tuning (RFT) (Yuan et al., 2023) and WizardMath (Luo et al., 2023), which fine-tunes LLMs using supervised data specific to certain datasets. Although such approaches improve in-domain performance, they cannot generalize to a wider range of math reasoning tasks beyond their fine-tuning data. For instance, both RFT and WizardMath can increase the accuracy on GSM8K (Cobbe et al., 2021) by 30%+, one of their fine-tuning datasets, but hurt the accuracy on out-of-domain datasets like MMLU-Math (Hendrycks et al., 2021a) or AQuA (Ling et al., 2017) by up to 10%.

In this paper, we aim to propose a lightweight yet generalizable math instruction-tuning approach to enhance the general (i.e., not limited to the fine-tuning tasks) mathematical reasoning capabilities of LLMs. Existing methods (Luo et al., 2023; Yuan et al., 2023; Taylor et al., 2022) primarily focus on Chain-of-Thought (CoT) approaches (Wei et al., 2022b; Nye et al., 2022) to solve math problems through step-by-step natural language descriptions. This approach excels in its generality to cover most math subjects but struggles with computation precision, and complex mathematical or algorithmic reasoning procedures (e.g., solving quadratic equation roots and calculating matrix eigenvalues). In contrast, prompts in the format of code like Program-of-Thought (PoT) approaches (Chen et al., 2022) and PAL (Madaan et al., 2022; Gao et al., 2023) utilize external tools (i.e., Python interpreter) to greatly simplify the math solving process. This approach advocates offloading the computation process to the external Python interpreter to solve complex mathematical and algorithmic reasoning procedures (e.g., solving quadratic equations with sympy or calculating matrix eigenvalues with numpy). However, PoT falls short in dealing with more abstract reasoning scenarios, like common-sense reasoning, formal logic, and abstract algebra, especially when there exist no built-in APIs.

To leverage the strengths of both CoT and PoT approaches, we introduce a new math hybrid instruction-tuning dataset `MathInstruct`, which has two main characteristics: (1) **broad coverage of different math fields and complexity levels**, and (2) **hybrid CoT & PoT rationales**. `MathInstruct` is based on seven existing math rationale datasets and six newly-curated datasets (see details in Table 1). We use `MathInstruct` to fine-tune Llama (Touvron et al., 2023a;b; Rozière et al., 2023) models of different scales ranging from 7B to 70B. The resulting `MAmmoTH` models ( Figure 1) demonstrate unprecedented potential in serving as math generalists.

We evaluate `MAmmoTH` on a spectrum of datasets, including in-domain (IND) test sets—GSM8K (Cobbe et al., 2021), MATH (Hendrycks et al., 2021b), AQuA-RAT (Ling et al., 2017), NumGLUE (Mishra et al., 2022b)—and out-of-domain (OOD) test sets—SVAMP (Patel et al., 2021), SAT (Zhong et al., 2023), MMLU-Math (Hendrycks et al., 2021a), Mathematics (Davies et al., 2021), and SimulEq (Koncel-Kedziorski et al., 2016). Compared with existing methods, our models generalize better to OOD datasets and substantially improve the performance of open-source LLMs in mathematical reasoning. Notably, on the popular competition-level MATH dataset (Hendrycks et al., 2021b), our 7B model can beat WizardMath (open-source MATH SoTA) (Luo et al., 2023) by 3.5x (35.2% vs 10.7%), and our 34B `MAmmoTH-Coder` (fine-tuned on Code Llama (Rozière et al., 2023)) can even beat the result of GPT-4 (using CoT).

We highlight our contributions from two perspectives: (1) From the **data engineering perspective**, we present `MathInstruct`, a high-quality math instruction tuning dataset, combining a variety of math problems and hybrid rationales. (2) From the **modeling perspective**, we investigate the impact of various data sources and input-output formats through training and evaluating over 50 different models and baselines ranging from 7B to 70B. Our models, including `MAmmoTH` and `MAmmoTH-Coder`, achieve substantial accuracy gains over existing open-source models.

| Training Dataset | Type | Annotation | # Samples | Characteristics | Fields |
|---|---|---|---|---|---|
| GSM8K (Cobbe et al., 2021) | CoT | Human | 7K | Grade Schol Exam | ■ |
| GSM8K-RFT (Yuan et al., 2023) | CoT | Llama | 28K | Llama + Validated | ■ |
| AQuA-RAT (Ling et al., 2017) | CoT | Human | 90K | GRE/GMAT Exam | ■ |
| MATH (Hendrycks et al., 2021b) | CoT | Human | 7K | Math Competition | ■ ■ ■ ■ ■ ■ ■ |
| TheoremQA (Chen et al., 2023) ✯ | CoT | GPT-4 | 600 | GPT4 + Validated | ■ ■ ■ ■ ■ |
| Camel-Math (Li et al., 2023a) | CoT | GPT-4 | 50K | GPT4 (Unvalidated) | ■ ■ ■ ■ ■ |
| College-Math ✯ | CoT | GPT-4 | 1.8K | GPT4 (Unvalidated) | ■ |
| GSM8K ✯ | PoT | GPT4 | 14K | GPT4 + Validated | ■ |
| AQuA-RAT ✯ | PoT | GPT4 | 9.7K | GPT4 + Validated | ■ |
| MATH ✯ | PoT | GPT4 | 7K | GPT4 + Validated | ■ ■ ■ ■ |
| TheoremQA ✯ | PoT | GPT4 | 700 | GPT4 + Validated | ■ ■ ■ ■ ■ |
| MathQA (Amini et al., 2019) | PoT | Human | 25K | AQuA-RAT Subset | ■ |
| NumGLUE (Mishra et al., 2022a) | PoT | Human | 13K | Lila Annotated | ■ |
| MathInstruct | | | 260K (72% CoT, 28% PoT) | | ■ ■ ■ ■ ■ ■ ■ |

Table 1: Overview of our MathInstruct. ✯ means with NEW rationales curated by us by prompting GPT-4. We have filtered out augmented samples that have answers inconsistent with the original dataset's annotations. Different colored squares represent different fields in mathematics: ■ Pre-Algebra; ■ Inter-Algebra; ■ Algebra; ■ Probability; ■ NumTheory; ■ Calculus; ■ Geometry.

## 2 OUR APPROACH

Mathematical reasoning serves as a vital gauge for assessing the ability of LLMs to execute complex multi-hop and quantitative reasoning. Previously, this has been a challenging task for neural networks, which struggle to solve even basic addition and subtraction problems (Yang et al., 2023). However, recent LLMs have considerable advancements in mathematical reasoning. Key breakthroughs have been made through CoT prompting (Wei et al., 2022b; Nye et al., 2022) and PoT prompting (Chen et al., 2022; Gao et al., 2023). CoT prompting encourages LLMs to solve problems incrementally on a scratchpad, enhancing both accuracy and explainability in mathematical reasoning. This approach contrasts with traditional methods that generate answers directly. PoT prompting, on the other hand, formulates the intermediate reasoning process as a program, executed with an external tool like Python, to compute the answer. This method improves robustness in solving complex mathematical problems by offloading the calculations to external tools. However, most existing work (Zhou et al., 2023a) in PoT is limited to proprietary models like GPT-4 (OpenAI, 2023) and Codex (Chen et al., 2021). The PoT potential of open-source models is yet to be seen. Our work aims at optimizing LLMs' CoT and PoT reasoning capabilities through instruction tuning.

### 2.1 CURATING A DIVERSE AND HYBRID INSTRUCTION TUNING DATASET

Our study aims to compile a list of high-quality and diverse math instruction-tuning datasets, standing out with two main characteristics: (1) broad coverage of different mathematical fields and complexity levels, and (2) hybrid CoT & PoT rationales.

**Broad Coverage of Different Math Fields and Complexity Levels:** We aim for a broad representation of math fields and complexity levels in our dataset. This ensures exposure to a diverse set of mathematical knowledge, fostering versatility in our models. Based on these criteria, we narrow down our choices to a few high-quality datasets that are widely adopted and encompass different math fields and complexity levels, such as GSM8K, MATH, AQuA, Camel, and TheoremQA. Furthermore, we notice a lack of coverage for college-level math knowledge, such as abstract algebra and formal logic, in existing datasets. To rectify this, we use GPT-4 to synthesize CoT rationales for questions in TheoremQA and create question-CoT pairs through Self-Instruct (Wang et al., 2023h), utilizing a few seed exemplars found online.

**Hybrid CoT and PoT Rationales:** Contrary to previous work (Yuan et al., 2023; Luo et al., 2023; Lee et al., 2023; Wang et al., 2023g) that focus on CoT, our dataset strategically combines both. This integration enhances the dataset's versatility, catering to varying mathematical problem-solving

approaches. However, most existing datasets provide limited program rationales, leading to an imbalance between CoT and PoT rationales. To fill the gap, we utilize GPT-4 to supplement the PoT rationales for selected datasets, including MATH, AQuA, GSM8K, and TheoremQA. We further enhance the dataset quality by meticulously removing near-duplicated solutions. These GPT-4 synthesized programs are then validated by comparing their executed results with ground truth, ensuring the high quality and reliability of the rationales. The validation subet ratio is shown in Table 6.

Following these guidelines, our instruction dataset, detailed in Table 1, encompasses 260K (instruction, response) pairs, covering a wide range of core mathematical fields (arithmetic, algebra, probability, calculus, and geometry, etc.), including hybrid CoT and PoT rationales, and offering diversity in both language and difficulty levels. This attests to its high quality and unique characteristics.

## 2.2 TRAINING SETUP

We unify all the subsets in our `MathInstruct` to conform to the structure of an Alpaca-like instruction dataset (Taori et al., 2023). This standardization ensures that the fine-tuned models can process data consistently, regardless of the original dataset formats. We choose the open-source models Llama-2 (Touvron et al., 2023b) and Code Llama (Rozière et al., 2023) as our base models. We fine-tune these models including 7B, 13B, 34B, and 70B on `MathInstruct`, which allows us to validate our `MathInstruct` at multiple scales. We fine-tune all the models with Huggingface transformers library (Wolf et al., 2019). We use a learning rate of 2e-5 for the 7B and 13B models, and 1e-5 for the 34B and 70B models. We set the batch size at 128 and used a cosine scheduler with a 3% warm-up period for three epochs. To efficiently train the computationally intensive 34B and 70B models, we employ DeepSpeed training with ZeRO-3 stage (Rajbhandari et al., 2020).

## 2.3 EVALUATION SETUP

Our hybrid training enables models to solve problems using either the CoT or PoT approach. By default, the model provides the CoT solution. To switch to the PoT approach, one can add the trigger phrase "Let's write a program to solve the problem" following the question.

Our preliminary evaluation reveals that PoT generally outperforms CoT, notably in open-form questions like GSM8K and MATH, as programmable solutions are better at solving complex mathematical and algorithmic reasoning procedures. However, PoT struggles with abstract reasoning scenarios such as commonsense reasoning, formal logic, and abstract algebra, particularly in the absence of built-in APIs. To further combine the power of both approaches, we introduce a simple hybrid decoding strategy: The model first attempts PoT prompting. If the program is not executable, we fall back to CoT prompting. This heuristic can further enhance our model's overall performance (see more discussions in Table 3.4). We also report the performance of self-consistency decoding method (Wang et al., 2023f) in Table 8.

## 3 EXPERIMENTS

### 3.1 EVALUATION DATASETS

We have selected diverse evaluation datasets (Table 2), encompassing a variety of **in-domain** and **out-of-domain** samples across diverse fields of mathematics, to assess the models' capabilities in general mathematical reasoning.

For the in-domain datasets, we consider GSM8K (Cobbe et al., 2021), MATH (Hendrycks et al., 2021b), AQuA-RAT (Ling et al., 2017), and NumGLUE (Mishra et al., 2022b). For the out-of-domain datasets, we choose SVAMP (Patel et al., 2021), Mathematics (Davies et al., 2021), SimulEq (Koncel-Kedziorski et al., 2016), SAT-Math (Zhong et al., 2023), and MMLU-Math (Hendrycks et al., 2021a). The wide selection of evaluation datasets includes math problems from elementary, high school, and college levels. Some of the datasets even include formal logic and commonsense reasoning. The choice of these datasets is to ensure a comprehensive evaluation of the models' capabilities to generalize to unfamiliar situations and different math fields. The chosen evaluation datasets consist of both open-formed questions and multi-choice questions.

| Eval Dataset | # Samples | In-Domain? | Answer Form | Fields |
|---|---|---|---|---|
| GSM8K (Cobbe et al., 2021) | 1319 | YES | Open-formed | ■ |
| MATH (Hendrycks et al., 2021b) | 5000 | YES | Open-formed | ■ ■ ■ ■ ■ ■ ■ |
| AQuA-RAT (Ling et al., 2017) | 254 | YES | Multi-choice | ■ |
| NumGLUE (Mishra et al., 2022b) | 1042 | YES | Open-formed | ■ |
| SVAMP (Patel et al., 2021) | 1000 | NO | Open-formed | ■ |
| Mathematics (Davies et al., 2021) | 1000 | NO | Open-formed | ■ ■ ■ ■ |
| SimulEq (Koncel-Kedziorski et al., 2016) | 514 | NO | Open-formed | ■ |
| SAT-Math (Zhong et al., 2023) | 220 | NO | Multi-choice | ■ ■ ■ |
| MMLU-Math (Hendrycks et al., 2021a) | 974 | NO | Multi-choice | ■ ■ ■ ■ |

Table 2: Comprehensive overview of our evaluation datasets, featuring a variety of in-domain and out-of-domain problems across diverse fields of mathematics. Different colored squares represent different fields in mathematics: ■ Pre-Algebra; ■ Inter-Algebra; ■ Algebra; ■ Probability; ■ NumTheory; ■ Calculus; ■ Geometry.

## 3.2 BASELINES

We partition our baselines into the following four categories:

- **Closed-source LLMs:** We consider 4 closed-source LLMs including GPT-4 (OpenAI, 2023), GPT-4 (Code Interpreter), PaLM-2 Unicorn (Anil et al., 2023), Claude-2 (Bai et al., 2022) and Codex (Chen et al., 2021). GPT-4, PaLM-2, and Claude-2 use CoT prompting while GPT-4 (Code Interpreter) and Codex use PoT prompting.
- **Llama Base:** For the base models, we consider Llama-1/2 (Touvron et al., 2023a;b), Llama-2-Chat (Touvron et al., 2023b).
- **Coder Model:** To compare with different coder models, we choose Code-Llama (Rozière et al., 2023), CodeT5+ (Wang et al., 2023i) and CodeGen (Nijkamp et al., 2023).
- **STEM Pre-training:** We cover Galactica (Taylor et al., 2022) mainly to understand the performance of models specialized in STEM knowledge.
- **Instruction Tuning:** We include Orca-Platypus (Mukherjee et al., 2023), Vicuna-1.5 (Zheng et al., 2023b), Tulu (Wang et al., 2023g), Platypus-2 (Lee et al., 2023) and Guanaco (Dettmers et al., 2023). We cover a wide spectrum of models trained with different types of datasets.
- **Dataset-Specific Tuning:** We include both RFT (Yuan et al., 2023) and WizardMath (Luo et al., 2023), which specifically tune the models to adapt to GSM8K and MATH datasets. We include them to understand their generalization.

For most baselines, we choose CoT prompting to maximize their performance due to their incompetence in program generation. All the 'Code Model' use PoT prompting. For GSM8K, MATH, AQuA, and NumGLUE, we will evaluate both 8-shot in-context-learning and zero-shot setups to report the highest score. For SVAMP, Mathematics, SimulEq, SAT, and MMLU, we use 5-shot in-context-learning to maintain consistency with prior work (Wei et al., 2022b; Chen et al., 2023). Our few-shot exemplars are mostly taken from PHP[1] (Zheng et al., 2023a). For MAmmoTH and MAmmoTH-Coder, we always evaluate under 0-shot setting. For all models, we allow a maximum sequence length of 2048 tokens for decoding. For multiple-choice questions, if the generated answer lacks an option, we map it by re-prompting the model: "Please find the closest option to [generated answer]. The options are [options]".

## 3.3 MAIN RESULTS

We report our in-domain and out-of-domain results in Table 3 and Table 4 respectively. Overall, we can see that MAmmoTH and MAmmoTH-Coder are able to outperform the SoTA model at different scales. In general, the performance gain for OOD datasets is more significant than IND datasets. These results show us the potential of our models as a mathematical generalist. On several datasets, MAmmoTH-Coder-34B and MAmmoTH-70B are even surpassing closed-source LLMs.

---

[1]https://github.com/chuanyang-Zheng/Progressive-Hint

| Model | Base | Math-SFT? | GSM8K | MATH | AQuA | NumGLUE | Avg |
|---|---|---|---|---|---|---|---|
| | | | Closed-source Model | | | | |
| GPT-4 | - | Unknown | 92.0[†] | 42.5[†] | 72.6[†] | 74.7 | 70.5 |
| GPT-4 (Code-Interpreter) | - | Unknown | 97.0[†] | 69.7[†] | - | - | - |
| PaLM-2 | - | Unknown | 80.7[†] | 34.3[†] | 64.1 | - | - |
| Claude-2 | - | Unknown | 85.2[†] | 32.5[†] | 60.9 | - | - |
| Codex (PoT) | - | No | 71.6[†] | 36.8[†] | 54.1[†] | - | - |
| ART (InstructGPT) | - | Unknown | 71.0 | - | 54.2 | - | - |
| | | | 7B Parameter Model | | | | |
| Llama-1 | - | No | 10.7[†] | 2.9[†] | 22.6 | 24.7 | 15.5 |
| Llama-2 | - | No | 14.6[†] | 2.5[†] | 30.3 | 29.9 | 19.3 |
| Galactica-6.7B | GAL | GAL-Instruct | 10.2 | 2.2 | 25.6 | 25.8 | 15.9 |
| Code-Llama (PoT) | - | No | 25.2 | 13.0 | 24.0 | 26.8 | 22.2 |
| AQuA-SFT | Llama-2 | AQuA | 11.2 | 3.6 | 35.6 | 12.2 | 15.6 |
| Llama-1 RFT | Llama-1 | GSM8K | 46.5[†] | 5.2 | 18.8 | 21.1 | 22.9 |
| WizardMath | Llama-2 | GSM8K+MATH | 54.9[†] | 10.7[†] | 26.3 | 36.1 | 32.0 |
| MAmmoTH | Llama-2 | MathInstruct | 53.6 | 31.5 | 44.5 | 61.2 | 47.7 |
| MAmmoTH-Coder | Code-Llama | MathInstruct | 59.4 | 33.4 | 47.2 | 66.4 | 51.6 |
| Δ | | | +5 | +21 | +12 | +30 | **+20** |
| | | | 13-15B Parameter Model | | | | |
| Llama-1 | - | No | 17.8[†] | 3.9[†] | 26.0 | 24.8 | 18.1 |
| Llama-2 | - | No | 28.7[†] | 3.9[†] | 25.1 | 8.8 | 16.6 |
| Code-Llama (PoT) | - | No | 36.1 | 16.4 | 28.7 | 29.2 | 27.6 |
| CodeT5+ (PoT) | - | No | 12.5 | 2.4 | 20.5 | 19.4 | 13.7 |
| CodeGen+ (PoT) | - | No | 12.7 | 3.4 | 24.5 | 22.5 | 15.7 |
| Vicuna-1.5 | Llama-2 | No | 28.4[†] | 5.8 | 24.8 | 36.9 | 23.9 |
| Llama-1 RFT | Llama-1 | GSM8K | 52.1[†] | 5.1 | 16.1 | 24.5 | 24.4 |
| Orca-Platypus | Llama-2 | Platypus | 38.4 | 3.0 | 18.9 | 35.3 | 23.9 |
| Platypus | Llama-2 | Platypus | 25.7 | 2.5 | 33.4 | 42.3 | 25.9 |
| WizardMath | Llama-2 | GSM8K+MATH | 63.9[†] | 14.0[†] | 21.2 | 40.8 | 34.9 |
| MAmmoTH | Llama-2 | MathInstruct | 62.0 | 34.2 | 51.6 | 68.7 | 54.1 |
| MAmmoTH-Coder | Code-Llama | MathInstruct | 64.7 | 36.3 | 46.9 | 66.8 | 53.7 |
| Δ | | | +1 | +20 | +18 | +26 | **+19** |
| | | | 30-34B Parameter Model | | | | |
| Llama-1 | - | No | 35.6[†] | 7.1[†] | 33.4 | 28.4 | 26.1 |
| Code-Llama (PoT) | - | No | 44.0 | 23.1 | 25.2 | 29.3 | 30.4 |
| Llama-1 RFT | Llama-1 | GSM8K | 56.5[†] | 7.4[†] | 18.5 | 24.3 | 26.6 |
| Galactica-30B | GAL | GAL-Instruct | 41.7 | 12.7 | 28.7 | 34.7 | 29.4 |
| Platypus | Llama-1 | Platypus | 37.8 | 9.3 | 27.9 | 40.5 | 28.8 |
| Tulu | Llama-2 | Tulu | 51.0 | 10.8 | 25.5 | 43.4 | 32.6 |
| MAmmoTH-Coder | Code-Llama | MathInstruct | 72.7 | 43.6 | 54.7 | 71.6 | 60.7 |
| Δ | | | +16 | +21 | +21 | +28 | **+28** |
| | | | 65-70B Parameter Model | | | | |
| Llama-1 | - | No | 50.9[†] | 10.6[†] | 35.0 | 50.2 | 36.6 |
| Llama-2 | - | No | 56.8[†] | 13.5[†] | 40.9 | 50.4 | 40.4 |
| Llama-2-Chat | Llama-2 | No | 54.9 | 18.6 | 37.0 | 51.6 | 40.5 |
| Guanaco | Llama-2 | No | 59.2 | 4.1 | 45.2 | 53.5 | 40.5 |
| WizardMath | Llama-2 | GSM8K+MATH | 81.6[†] | 22.7[†] | 20.0 | 48.9 | 43.3 |
| Platypus | Llama-2 | Platypus | 70.6 | 15.6 | 51.2 | 55.4 | 48.1 |
| MAmmoTH | Llama-2 | MathInstruct | 76.9 | 41.8 | 65.0 | 74.4 | 64.5 |
| Δ | | | -5 | +19 | +14 | +19 | **+16** |

Table 3: The table compiles all the **in-domain** evaluation results. Results marked as † are copied from other papers, which can be found on paperswithcode leaderboards. Math-SFT? means whether the model has been instruction-tuned on any math reasoning datasets. Pink numbers highlight the highest number within the corresponding scale and dataset. Note that there does not exist a 30B+ version for Llama-2 or a 70B version for Code-Llama.

From Table 3, we can observe that our main competitors for IND datasets are WizardMath (Luo et al., 2023) and Platypus (Lee et al., 2023). WizardMath's training is heavily rooted in GSM8K

| Model | SVAMP | Mathematics | SimulEq | SAT-Math | MMLU-Math | Avg |
|---|---|---|---|---|---|---|
| *Closed-source Model* | | | | | | |
| GPT-4 | 97.0[†] | 60.8 | 83.1 | 95[†] | 74.9 | 82.1 |
| Codex (PoT) | 85.2[†] | - | - | 68[†] | - | - |
| *7B Parameter Model* | | | | | | |
| Llama-1 | 24.5 | 6.2 | 4.6 | 22.7 | 30.6 | 17.7 |
| Llama-2 | 34.5 | 6.0 | 5.0 | 26.8 | 29.8 | 20.4 |
| Code-Llama (PoT) | 49.4 | 21.7 | 3.5 | 28.6 | 26.9 | 26.0 |
| Llama-1 RFT | 21.1 | 5.1 | 11.0 | 12.5 | 21.7 | 14.3 |
| Galactica-6.7B | 25.6 | 4.6 | 4.2 | 17.5 | 28.0 | 16.0 |
| WizardMath | 36.1 | 9.3 | 12.8 | 25.4 | 31.1 | 28.6 |
| Toolformer | 29.4[†] | - | - | - | - | - |
| MAmmoTH | 67.7 | 46.3 | 41.2 | 42.7 | 42.6 | 48.1 |
| MAmmoTH-Coder | 71.4 | 55.4 | 45.9 | 40.5 | 48.3 | 52.3 |
| Δ | +22 | +34 | +33 | +14 | +17 | **+24** |
| *13B Parameter Model* | | | | | | |
| Llama-1 | 34.7 | 6.9 | 5.4 | 27.7 | 30.7 | 21.0 |
| Llama-2 | 35.1 | 11.5 | 5.8 | 32.7 | 34.4 | 23.9 |
| Code-Llama (PoT) | 60.0 | 21.3 | 3.8 | 25.9 | 27.7 | 27.7 |
| Vicuna-1.5 | 55.7 | 10 | 6.6 | 34.0 | 34.1 | 28.1 |
| Llama-1 RFT | 46.5 | 6.7 | 10.1 | 13.2 | 21.6 | 19.6 |
| WizardMath | 51.9 | 14.1 | 14.9 | 24.5 | 32.1 | 27.5 |
| Platypus | 55.4 | 11.4 | 7.4 | 36.8 | 35.5 | 29.3 |
| Orca-Platypus | 56.8 | 12.6 | 7.9 | 29.5 | 41.6 | 29.7 |
| MAmmoTH | 72.4 | 49.2 | 43.2 | 46.8 | 47.6 | 51.8 |
| MAmmoTH-Coder | 73.7 | 61.5 | 47.1 | 48.6 | 48.3 | 55.8 |
| Δ | +14 | +40 | +33 | +12 | +7 | **+26** |
| *30-34B Parameter Model* | | | | | | |
| Llama-1 | 48.8 | 12.8 | 11.2 | 33.4 | 39.0 | 29.0 |
| Code-Llama (PoT) | 69.1 | 34.5 | 6.8 | 26.8 | 21.6 | 31.7 |
| Llama-1 RFT | 55.4 | 7.6 | 12.8 | 20.4 | 37.9 | 26.8 |
| Galactica-30B | 41.6 | 11.8 | 13.2 | 37.7 | 37.9 | 28.4 |
| Tulu | 59.0 | 10.7 | 10.3 | 31.3 | 39.8 | 30.2 |
| Platypus | 51.7 | 13.8 | 13.6 | 38.6 | 41.0 | 31.7 |
| MAmmoTH-Coder | 84.3 | 65.4 | 51.8 | 60.9 | 53.8 | 63.2 |
| Δ | +15 | +31 | +38 | +22 | +13 | **+32** |
| *65-70B Parameter Model* | | | | | | |
| Llama-1 | 55.3 | 14.2 | 15.2 | 37.4 | 44.1 | 33.2 |
| Llama-2 | 63.8 | 20.5 | 14.0 | 51.3 | 47.1 | 39.3 |
| Llama-2-Chat | 71.5 | 19.2 | 21.7 | 44.1 | 46.9 | 40.6 |
| WizardMath | 71.8 | 17.1 | 37.9 | 13.2 | 27.4 | 33.4 |
| Guanaco | 66.8 | 17.8 | 20.2 | 50.0 | 47.3 | 40.4 |
| Platypus | 51.8 | 26.3 | 21.7 | 55.9 | 52.5 | 41.6 |
| MAmmoTH | 82.4 | 55.6 | 51.4 | 66.4 | 56.7 | 62.5 |
| Δ | +11 | +29 | +14 | +11 | +4 | **+21** |

Table 4: The table compiles all the **out-of-domain** evaluation results. Results marked as † are copied from other papers, which can be found on paperswithcode leaderboards.

and MATH datasets. Therefore, WizardMath's results are highly competitive on these two datasets. However, the dataset-specific training can be detrimental to OOD datasets like AQuA. In contrast, Platypus fine-tunes LLMs on a wide range of text and math reasoning datasets. it improves the open-source SoTA on several datasets. Similarly, MAmmoTH can achieve universal improvement across the board. A major observation is that MAmmoTH is particularly strong at solving more complex math problems in MATH, where the gain of our model over WizardMath (open-source SoTA on MATH) can exceed 25% at different scales.

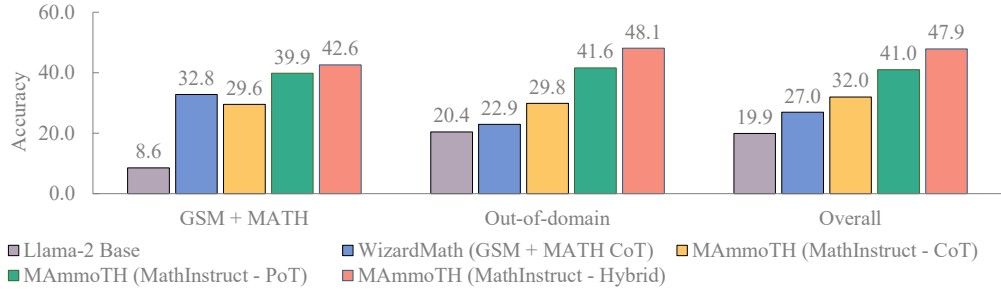

Figure 2: Investigation of the influence of CoT & PoT hybrid training on the 7B Llama-2 model. "Out-of-domain" refers to the five datasets detailed in Table 2. Key insights include: 1) The SoTA model, utilizing dataset-specific CoT fine-tuning on GSM and MATH, displays strong performance within its domains but struggles in OOD scenarios; 2) Diverse data sources in `MathInstruct` enable better math generalist model; 3) Fine-tuning on the PoT subsets generally outperforms fine-tuning on the CoT subsets; 4) Hybrid training yields the best-performing model. The breakdown results on each dataset can be found in Appendix Table 9.

From Table 4, we can observe that our main competitor for OOD datasets is Platypus (Lee et al., 2023). Similar to in-domain results, Platypus is able to yield gains over the baseline models universally across the board, especially on the MMLU-Math dataset, which is tied with `MAmmoTH`-70B. It is worth noting that the performance gains of our model on OOD datasets are even more significant than on in-domain datasets. This demonstrates our models' remarkable generalizability to unseen math problems. Notably, `MAmmoTH`-7B also boosts the CoT performance of WizardMath-7B greatly on MMLU-Math by 9%, which contains a substantial number of questions beyond the subjects we covered in our training dataset.

**Comparison between Different Base Models.** In our experiments, we experimented with both Llama-2 and Code-Llama as the base models. From the two tables, we can observe that Code-Llama is consistently better than Llama-2, especially on OOD datasets. The gap between `MAmmoTH` and `MAmmoTH-Coder` can even reach up to 5%. Surprisingly, the average performance on OOD datasets of `MAmmoTH-Coder` (34B) is actually higher than `MAmmoTH` (70B). We believe `MAmmoTH-Coder` benefits greatly from the continuous code training of Code-Llama, which not only enhances the PoT capabilities but also improves Llama's general reasoning skills.

### 3.4 ABLATION STUDY ON DATA SOURCE

**Ablation of the Data Source.** In order to better understand what factors contribute to the great gain of `MAmmoTH` over existing baselines, we set up a group of control experiments in Figure 2. We study the following setups:

(1) `MAmmoTH` (`MathInstruct`- CoT): This experiment aims to understand how much our curated CoT data could improve the generalization over the SoTA model WizardMath (Luo et al., 2023) trained specifically on GSM + MATH. As can be seen, while sacrificing accuracy on GSM + MATH by 3%, our CoT subset fine-tuning improves the overall nine-dataset accuracy from 27% to 32%.

(2) `MAmmoTH` (`MathInstruct`- PoT): This experiment aims to understand the advantage of our PoT subset. As can be observed, our PoT subset fine-tuning can significantly improve the overall accuracy from 27% to 41%. This ablation reflects the importance of unlocking the program generation capabilities of our model.

(3) `MAmmoTH` (`MathInstruct`- Hybrid): We further combine CoT and PoT as the hybrid training data to achieve the best overall performance of 47.9%. This combined gain comes from two aspects:

- The CoT subset helps maintain generic language-based reasoning skills to handle scenarios where PoT cannot handle well, e.g., abstract reasoning multi-choice questions in AQuA and MMLU.

- The PoT subset can teach the model how to utilize Python APIs to solve complex math problems with high precision, e.g., the MATH problems requiring complex computation.

| Training Data | GSM | MATH | AQuA | NumG | SVA | Mat | Sim | SAT | MMLU | AVG |
|---|---|---|---|---|---|---|---|---|---|---|
| - | 14.6 | 2.5 | 30.3 | 29.9 | 34.5 | 6.0 | 5.0 | 26.8 | 29.8 | -25.3 |
| G | 56.6 | 9.2 | 24.4 | 32.1 | 65.4 | 20.5 | 12.3 | 27.2 | 25.2 | 30.3 |
| M | 27.1 | 25.5 | 27.8 | 32.1 | 47 | 49.4 | 10.5 | 26.4 | 27.4 | 30.4 |
| A | 15.3 | 5.8 | 39.7 | 15.5 | 15.3 | 6.3 | 7.2 | 32.7 | 36.6 | 19.4 |
| G + M | 58.1 | 28.2 | 26.0 | 34.7 | 64.8 | 50.1 | 17.1 | 28.6 | 28.4 | 37.3 |
| G + M + T | 56.5 | 26.5 | 27.4 | 35.5 | 64.4 | 50.6 | 18.8 | 29.1 | 29.1 | 37.5 |
| G + M + C | 57.4 | 28.5 | 26.2 | 37.5 | 65.3 | 50.4 | 17.7 | 29.3 | 28.7 | 37.9 |
| G + M + A | 56.1 | 27.1 | 37.8 | 37.2 | 64.8 | 48.2 | 19.8 | 35.4 | 39.8 | 40.7 |
| G + M + C + A | 57.5 | 29.1 | 46.9 | 42.2 | 65.8 | 49.6 | 32.7 | 42.3 | 43.1 | 45.5 |
| M + C + A + N | 24.7 | 26.1 | 39.4 | 59.7 | 61.6 | 48.6 | 43.4 | 36.4 | 41.2 | 42.3 |
| G + M + C + N | 50.8 | 26.2 | 20.9 | 65.5 | 65.8 | 48.5 | 41.4 | 26.4 | 24.7 | 41.1 |
| G + C + A + N | 51.3 | 14.8 | 41.7 | 58.8 | 66.3 | 31.0 | 42.2 | 34.1 | 40.5 | 42.3 |
| G + M + C + A + T | 55.4 | 28.6 | 42.5 | 44.9 | 65.4 | 50.8 | 34.9 | 41.3 | 42.5 | 45.1 |
| G + M + C + A + N | 56.5 | 28.9 | 38.2 | 63.7 | 64.1 | 47.9 | 40.8 | 38.6 | 44.5 | 47.0 |
| G + M + C + A + N +T | 53.8 | 27.0 | 38.2 | 60.8 | 65.9 | 50.8 | 41.8 | 42.5 | 42.7 | 47.1 |
| G + M + C + A + N + MQA | 55.7 | 28.8 | 42.5 | 62.1 | 64.6 | 45.9 | 38.9 | 41.3 | 45.0 | 47.2 |
| `MathInstruct` | 53.6 | 31.5 | 44.5 | 61.2 | 67.7 | 46.3 | 41.2 | 42.7 | 42.6 | 47.9 |

Table 5: Influence of different major subsets in `MathInstruct` based on Llama-2 7B. G: GSM8K, M: MATH, C: Camel, A: AQuA, N: NumGLUE, MQA: MathQA. "Existing data": the subset of `MathInstruct` in Table 1 by excluding all the NEW rationales curated by us. We shorten Mathematics as Mat, SimulEq as Sim, NumGLUE as NumG, and SVAMP as SVA to save space.

We put some case studies in Appendix B to demonstrate the respective advantages of PoT and CoT in solving different types of math problems. To summarize, we attribute our substantial gain to: 1) diverse data sources covering different math fields and complexity levels and 2) a hybrid of CoT & PoT instruction tuning and decoding strategy.

**Influence of Major Subsets.** Given the diverse sources of `MathInstruct` used in training `MAmmoTH`, it is important to understand how each dataset contributes to the overall performance of the model. We focus on five significant subsets: GSM8K, MATH, Camel, AQuA, and NumGLUE. We conduct an experiment gradually adding each dataset into training and compare the performance with the one fine-tuned on the whole `MathInstruct`. These results underscore the significant impact of diverse data sources on `MAmmoTH` performance, a core aspect of making `MAmmoTH` a math generalist. The results also provide valuable insights for future data curation and collection efforts (e.g., we should always collect diverse data and avoid collecting only specific types of data).

To help understand the contribution of the 6 newly curated datasets as shown in Table 1, we remove them from `MathInstruct`, and train a model on the existing data. As shown in the last two rows of Table 5, our new curated data substantially improves the performance on many datasets and leads to a 9% overall increase, which reflects the importance of the NEWLY curated dataset.

**Influence of Hybrid Decoding.** To demonstrate the effectiveness of the hybrid decoding method, we conduct an experiment as outlined in subsection 2.3. By default, we initially attempt the PoT decoding method for a given question. If it fails to generate an executable query, we then transition to the CoT decoding method. The performance of different decoding methods (CoT, PoT, and Hybrid) is shown in Table 10. This hybrid decoding improves performance on every test set, showcasing that our model can effectively leverage the strengths of both CoT and PoT decoding strategies.

## 4 CONCLUSION

In this paper, we propose a novel math instruction tuning approach to activate open-source LLMs' mathematical reasoning capabilities. Through a comprehensive study, we show that our models can outperform the SoTA performance at different scales by a huge margin. Our models benefit massively from: 1) the broad coverage of different math fields and complexity levels, and 2) a hybrid of CoT and PoT training. Our instruction tuning dataset contains 260K samples, which makes fine-tuning highly affordable even for academic labs. Our work paves the road for future studies to activate LLMs' core capabilities in specialized domains.

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

# A    RELATED WORK

## A.1    MATHEMATICAL REASONING DATASETS

Our work builds upon the existing mathematical reasoning literature. Early on, mathematical reasoning is mostly focused on solving synthetic basic math problems like AddSub (Hosseini et al., 2014) and other arithmetic reasoning datasets (Koncel-Kedziorski et al., 2015; Roy & Roth, 2015; Patel et al., 2021). Later on, more difficult math word problem datasets (Cobbe et al., 2021; Amini et al., 2019; Ling et al., 2017; Hendrycks et al., 2021b) have been proposed to focus on addressing realistic math word problems. NumGLUE (Mishra et al., 2022b) and LiLA (Mishra et al., 2022a) compile the existing literature to build a more diversified dataset collection. However, these datasets are mostly focused on grade school math problems. To further test LLMs' limits in addressing more complex math problems, MMLU (Hendrycks et al., 2021a) includes college math problems in its evaluation suite. More recently, (Chen et al., 2023; Wang et al., 2023e) have proposed to tackle more challenging college-level science and math problems. Our instruction tuning dataset is built upon existing work to include a diversified collection of math problems from different subfields.

## A.2    REASONING WITH LARGE LANGUAGE MODELS

LLMs have demonstrated great capabilities to reason with the help of Chain-of-Thought prompting (Wei et al., 2022b; Kojima et al., 2022; Wang et al., 2023f). Suzgun et al. (2022) have shown that CoT can already surpass human performance on challenging BIG-Bench tasks. Later on, several other works (Drozdov et al., 2023; Zhou et al., 2023c; Nye et al., 2022; Wang et al., 2022a; 2023a; Li et al., 2023b; Wang et al., 2023d; Yu et al., 2023) also propose different approaches to utilize LLMs to solve reasoning tasks by allowing intermediate steps. ReAct Yao et al. (2023) proposes to leverage external tools like search engines to enhance LLM reasoning skills. Another trend is to enable LLMs' capabilities to use programs as thought processes like PoT (Chen et al., 2022). Some follow-up works include self-critic (Gou et al., 2023), self-eval (Xie et al., 2023), plan-and-solve (Wang et al., 2023c). These methods propose to enhance LLMs' capabilities to solve math problems with PoT. Self-critic (Gou et al., 2023) and self-eval (Xie et al., 2022) both adopt self-evaluation to enhance the robustness of the generated program. Plan-and-solve (Wang et al., 2023c) instead adopts more detailed planning instructions to help LLMs create a high-level reasoning plan. These methods all prove to bring decent improvements over PoT.

## A.3    INSTRUCTION TUNING IN LANGUAGE MODELS

Instruction tuning is part of a line of work designed to "align" language models with more useful objectives and human preferences. The instruction tuning step is seen as a major step to activate LLMs' certain capabilities to respond to human instructions. Previously, instruction tuning is mainly focused on enhancing LLMs' general-purpose instruction following abilities. Since 2021, CrossFit (Ye et al., 2021) and NaturalInstruction (Wang et al., 2022b), FLAN (Wei et al., 2022a) and T0 (Sanh et al., 2022) are amongst the first wave of instruction tuning effort to understand LLMs' generalization capabilities. Later on, FLAN-v2 (Chung et al., 2022; Longpre et al., 2023) have been proposed to understand the effect of scaling up the instruction datasets to understand its impact on model performance. These approaches mainly adopt human-annotated datasets to build the instruction following dataset. More recently, multiple works (Wang et al., 2023h; Xu et al., 2023; Peng et al., 2023; Zhou et al., 2023b; Wang et al., 2023g) propose to utilize synthetic instruction following data distilled from GPT-3/4 to align open-source LLMs. The most similar effort to ours is Platypus (Lee et al., 2023) which aims to utilize a domain-specialized dataset to construct a small-scale instruction following dataset to enhance LLMs' reasoning capabilities.

## B  CASE STUDY

We conduct a comparison between our PoT results vs. CoT results in Figure 3, Figure 4 and Figure 5. In the first example, even though PoT and CoT can both solve the problem, CoT gives a very tedious solution to derive the answer. Such solution is not only slow but also unstable. In the second and third case, we can further see the advantages of PoT over CoT by utilizing external tools and Python packages to greatly simplify the solution. Figure 6 shows some types of questions (especially the formal logic question) that are not easily handled by programs. In order to address these types of questions, CoT is a better choice.

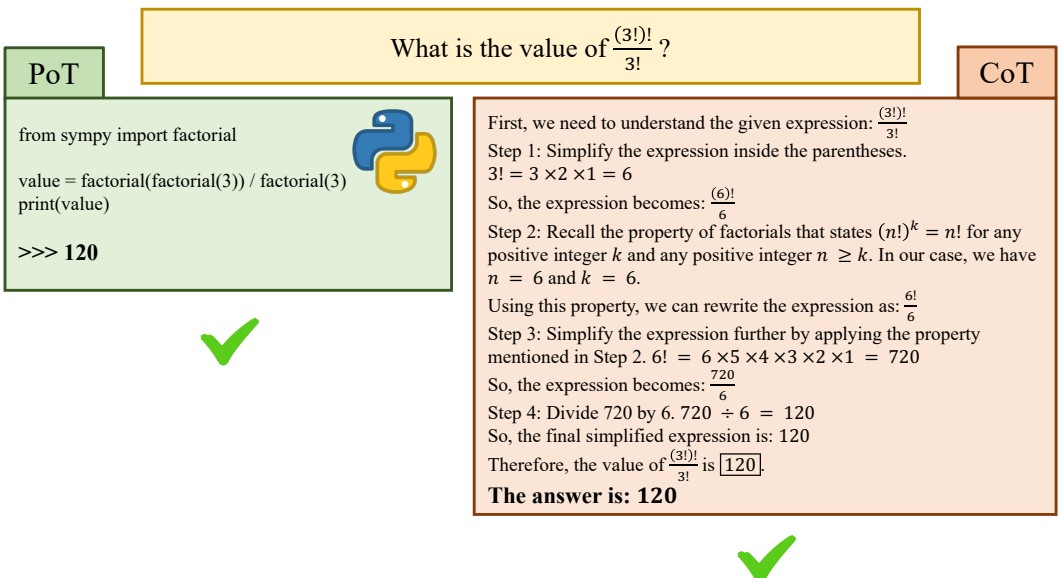

Figure 3: Example 1: PoT and CoT can both solve the problem, however, CoT gives a very tedious solution to derive the answer.

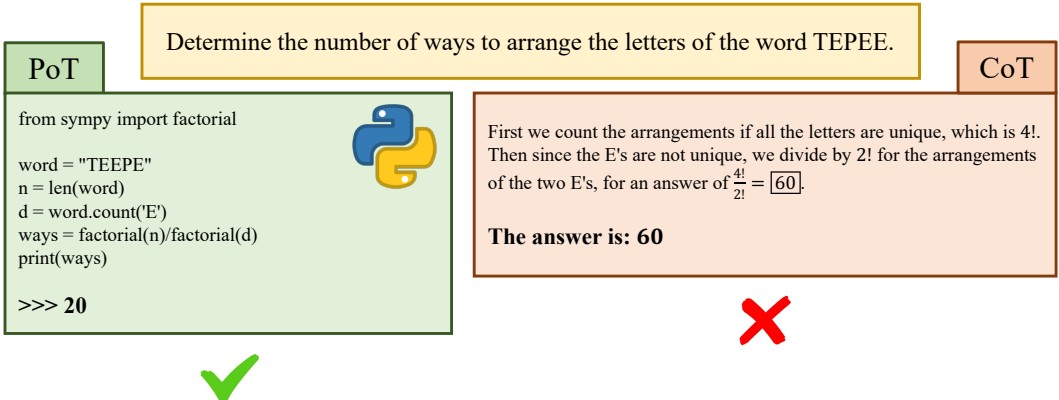

Figure 4: Example 2: PoT generates the correct solution while CoT fails.

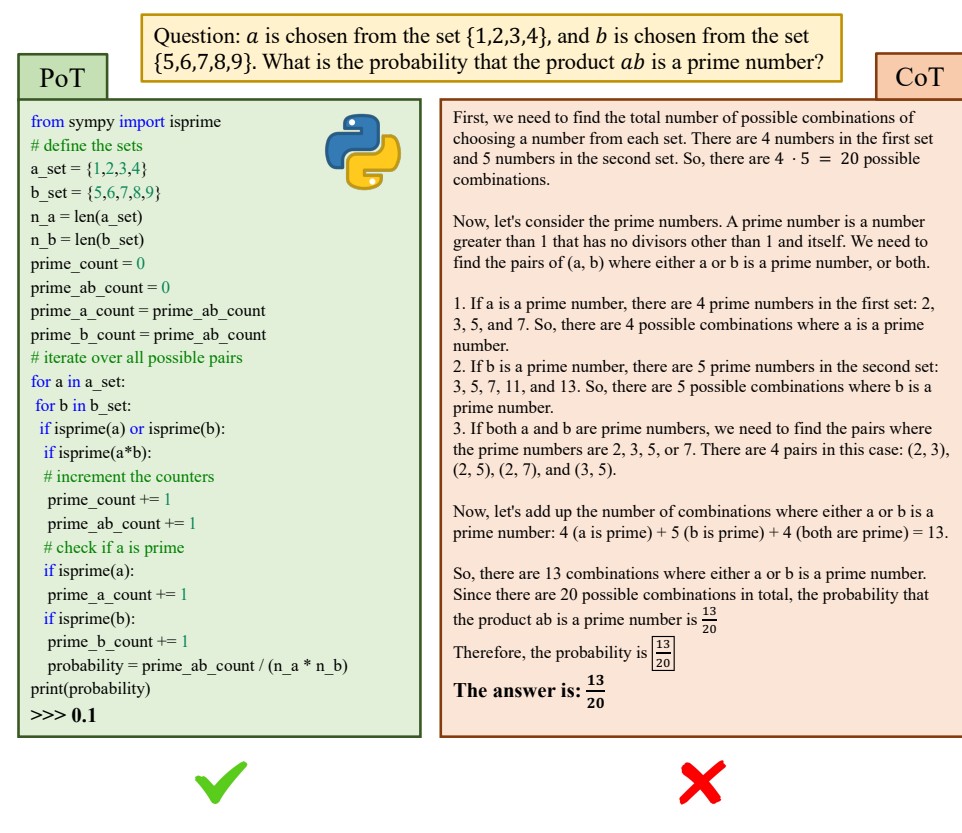

Figure 5: Example 3: PoT generates the correct solution while CoT fails.

---

If $n$ is such that $36 \leq n \leq 72$, then $x = (n2 + 2\sqrt{n}(n + 4) + 16)/(n + 4\sqrt{n} + 4)$ satisfies.
Answer Choices: (A) $20 < x < 54$ (B) $23 < x < 58$ (C) $25 < x < 64$ (D) $28 < x < 60$ (E) None of these

---

The expression in the numerator can be rewritten as $(n + 2\sqrt{n} + 4)^2$, and so the expression is equal to $n + 2\sqrt{n} + 4$. We substitute the smallest value of $n$, which is 36, and get 42. We substitute the largest value of $n$, which is 72, and get 88. **So the answer is (E) None of these**

CoT

---

Which of the following could NOT be the sides of an isosceles triangle?
Answer Choices: (A) 2,2,3 (B) 2,2,4 (C) 4,12,12 (D) 5,5,5

---

This is because in an isosceles triangle, the lengths of two sides are equal, and the length of the third side must be less than the sum of the lengths of the other two sides. In this case, 4 is equal to the sum of the other two sides (2+2), not less. **Therefore, the answer is (B) 2,2,4.**

CoT

---

Select the best translation into predicate logic. Some CEO is wealthier than everyone. ($Cx$: $x$ is a CEO; $Px$: $x$ is a person; $Wxy$: $x$ is wealthier than $y$)
Answer Choices: (A) $(\exists x)[Cx \bullet (\forall y)(Py \bullet Wxy)]$ (B) $(\exists x)[Cx \supset (\forall y)(Py \supset Wxy)]$
(C) $(\exists x)[Cx \bullet (\forall y)(Py \supset Wxy)]$ (D) $(\forall x)[Px \supset (\exists y)(Cy \supset Wxy)]$

---

$\exists x$: There exists some $x$ $Cx$: $x$ is a CEO, and
$\forall y$: For all $y$ $Py$: $y$ is a person, and $Wxy$: $x$ is wealthier than $y$
So, in plain English, this expression says: "There exists some $x$ such that $x$ is a CEO and for all $y$, if $y$ is a person then $x$ is wealthier than $y$." Which matches the original statement.
**Therefore the answer is (A) $(\exists x)[Cx \bullet (\forall y)(Py \bullet Wxy)]$**

CoT

Figure 6: Example 4: Some types of questions (e.g., formal logic) are hard to be solved by PoT but could be handled by CoT.

## C  KEY INSIGHTS

We show the results in Table 5. And our key findings are summarized as follows:

- When the model is trained on individual datasets (GSM8K, MATH, or AQuA), it demonstrates strong performance in tasks closely related to the respective training dataset. However, it generally struggles to generalize this performance to other tasks.

- The datasets GSM8K, MATH, and AQuA emerge as crucial components in the training process. Training the model on a combination of these datasets leads to a marked improvement in its overall performance, indicative of the complementary nature of these datasets. Omitting any one of these datasets results in a noticeable decline in overall performance.

- TheoremQA, despite its smaller scale, exerts some influence on the model's performance. It tends to enhance the model's capabilities on datasets like Mathematics and SimulEq, yet it may slightly impede performance on GSM8K and MATH. Nevertheless, its inclusion in the training regimen is favored, as it introduces a broader variety of questions and could potentially aid the model in generalizing to other complex problems that require advanced computational skills.

- Incorporating a wider range of datasets into the training consistently improves the model's performance across most tasks. This suggests that a broad and diverse knowledge base is instrumental in enhancing the model's ability to generalize.

# D ADDITIONAL RESULTS

We added some additional results here. The results provides how much of GPT-4 generated data are being filtered (see Table 6). The results also include the generalization benefits of MathInstruct vs. only training on GSM (see Table 7). We also include different self-consistency setup to show how far our method can reach with ensemble decoding methods.

|  | GPT-4 Rationales Validated Ratio |
| --- | --- |
| AQuA-PoT | 75% (9772 /13000) |
| MATH-PoT | 51% (7088 / 14000) |
| TheoremQA-PoT | 44% (703 / 1600) |
| TheoremQA-CoT | 37% (592 / 1600) |
| GSM8K - PoT | 81% (14591 / 18000) |

Table 6: The validated ratio of GPT-4 distilled rationales of newly curated subsets in Table 1.

|  | GSM | MATH | AQUA | NumG | SVA | Mat | Sim | SAT | MMLU | AVG |
| --- | --- | --- | --- | --- | --- | --- | --- | --- | --- | --- |
| CodeLlama - 7B - GSM8K | **63.2** | 18.4 | 25.4 | 32.9 | 67.6 | 40.1 | 16.3 | 26.4 | 26.1 | 35.2 |
| CodeLlama - 7B - MathInstruct | 59.4 | **33.4** | **47.2** | **66.4** | **71.4** | **55.4** | **45.9** | **40.5** | **48.3** | **52.0** |

Table 7: CodeLlama 7B trained on GSM8K and `MathInstruct`. Training exclusively on a single dataset like GSM8K leads to challenges in generalizing to other datasets

| Decoding Method | GSM | MATH |
| --- | --- | --- |
| 1 PoT | 58.8 | 32.1 |
| 1 PoT + 1 CoT backup | 59.4 | 33.4 |
| 5 PoT + 5 CoT self-consistency | 66.4 | 37.0 |
| 10 PoT self-consistency | 67.0 | 37.8 |
| 10 PoT + 10 CoT self-consistency | 69.6 | 38.2 |
| 20 PoT self-consistency | 70.1 | 38.6 |

Table 8: Exploration of different decoding methods. Though the self-consistency Wang et al. (2023f) decoding can further improve the performance, it introduces significantly larger inference. Our chosen strategy, while straightforward, strikes an optimal balance between complexity, performance, and practical applicability.

| Model | GSM | MATH | AQuA | NumG | SVA | Mat | Sim | SAT | MMLU | Overall |
|---|---|---|---|---|---|---|---|---|---|---|
| Base | 14.6 | 2.5 | 30.3 | 29.9 | 34.5 | 6.0 | 5.0 | 26.8 | 29.8 | 19.9 |
| WizzardMath | 54.9 | 10.7 | 26.3 | 36.1 | 36.1 | 9.3 | 12.8 | 25.4 | 31.1 | 27.0 |
| MAmmoTH (MathInstruct-CoT) | 49.2 | 9.9 | 42.2 | 37.1 | 48.5 | 9.5 | 17.3 | 34.1 | 39.8 | 32.0 |
| MAmmoTH (MathInstruct-PoT) | 50.8 | 28.9 | 28.6 | 52.7 | 65.0 | 46.7 | 42.0 | 25.9 | 28.3 | 41.0 |
| MAmmoTH (MathInstruct) | 53.6 | 31.5 | 44.5 | 61.2 | 67.7 | 46.3 | 41.2 | 42.7 | 42.6 | 47.9 |

Table 9: Breakdown results of Figure 2. Investigation of the influence of CoT & PoT hybrid training on the 7B Llama-2 model.

| Model | Decoding | GSM | MATH | AQuA | NumG | SVA | Mat | Sim | SAT | MMLU | AVG |
|---|---|---|---|---|---|---|---|---|---|---|---|
| MAmmoTH-7B | CoT | 50.5 | 10.4 | 43.7 | 44.0 | 47.3 | 9.2 | 18.9 | 32.7 | 39.9 | 33.0 |
| | PoT | 51.6 | 28.7 | 43.3 | 52.3 | 65.1 | 41.9 | 48.2 | 39.1 | 44.6 | 46.1 |
| | **Hybrid** | **53.6** | **31.5** | **44.5** | **61.2** | **67.7** | **46.3** | **41.2** | **42.7** | **42.6** | **47.9** |
| MAmmoTH-Coder-7B | CoT | 22.4 | 7.9 | 36.2 | 36.0 | 37.0 | 8.2 | 7.2 | 32.7 | 34.6 | 24.7 |
| | PoT | 58.8 | 32.1 | 47.2 | 57.1 | 71.1 | 53.9 | 44.6 | 40.0 | 47.8 | 50.3 |
| | **Hybrid** | **59.4** | **33.4** | **47.2** | **66.4** | **71.4** | **55.4** | **45.9** | **40.5** | **48.3** | **52.0** |
| MAmmoTH-13B | CoT | 56.3 | 12.9 | 45.3 | 45.6 | 53.8 | 11.7 | 22.4 | 43.6 | 42.3 | 37.1 |
| | PoT | 61.3 | 32.6 | 48.8 | 59.6 | 72.2 | 48.5 | 40.3 | 46.8 | 45.4 | 50.6 |
| | **Hybrid** | **62.0** | **34.2** | **51.6** | **68.7** | **72.4** | **49.2** | **43.2** | **46.8** | **47.6** | **52.9** |
| MAmmoTH-Coder-13B | CoT | 32.1 | 10.2 | 40.6 | 36.2 | 43.0 | 9.6 | 10.1 | 40.9 | 36.6 | 28.8 |
| | PoT | 64.3 | 35.2 | 46.8 | 54.2 | 73.2 | 60.0 | 44.2 | 48.2 | 48.2 | 52.7 |
| | **Hybrid** | **64.7** | **36.3** | **46.9** | **66.8** | **73.7** | **61.5** | **47.1** | **48.6** | **48.3** | **54.9** |
| MAmmoTH-Coder-34B | CoT | 34.3 | 11.6 | 39.0 | 36.2 | 44.6 | 10.8 | 10.9 | 46.4 | 42.9 | 30.7 |
| | PoT | 72.3 | 42.8 | 53.8 | 59.6 | 84.0 | 64.7 | 50.6 | 58.6 | 52.7 | 59.9 |
| | **Hybrid** | **72.7** | **43.6** | **54.7** | **71.6** | **84.3** | **65.4** | **51.8** | **60.9** | **53.8** | **62.1** |
| MAmmoTH-70B | CoT | 72.4 | 21.1 | 57.9 | 58.9 | 71.6 | 20.0 | 31.9 | 57.3 | 52.1 | 49.2 |
| | PoT | 76.7 | 40.1 | 60.2 | 64.3 | 81.7 | 55.3 | 45.3 | 64.1 | 53.5 | 60.1 |
| | **Hybrid** | **76.9** | **41.8** | **65.0** | **74.4** | **82.4** | **55.6** | **51.4** | **66.4** | **56.7** | **63.4** |

Table 10: Influence of different decoding methods on each dataset.

# E    LIMITATIONS

Despite their training on a diverse set of mathematical rationale datasets, the `MAmmoTH` models might exhibit limitations when faced with problems outside their primary domain of expertise like mathematical analysis, complex analysis, graph theory, numerical analysis, etc. Thus, our models are not suitable for solving more complex problems in these fields. Also, they have not been trained with proof-type problems, thus their theorem-proving capability is also limited. In the future, we would like to expand the models' skill set to cover more fields and theorem-proving problems.

There is also a risk of the `MAmmoTH` models generating potentially harmful, offensive, or biased content, especially if they are asked to answer questions beyond math. The `MAmmoTH` series could be misused for malicious purposes, such as spreading misinformation or probing sensitive topics. Developers should conduct safety testing and tuning tailored to their specific applications before deploying any `MAmmoTH` model. While we have made every effort to ensure the cleanliness and purity of our training data, we cannot guarantee absolute perfection. It is unlikely but not impossible that some inappropriate questions slipped through the curation process.

Future work may continue to explore how to further improve the robustness and generalizability of `MAmmoTH` in mathematical reasoning. For example, recent work identifies "sycophancy" and "Clever Hans effect" in reasoning: LLMs cannot maintain truthful solutions to reasoning tasks when challenged by the user's absurdly invalid arguments and critiques (Wang et al., 2023b). Potential methods to improve the models' reasoning robustness could involve the exploration of synthetic data intervention methods as explored in (Wei et al., 2023).

