# OpenReview forum: "MAmmoTH: Building Math Generalist Models through Hybrid Instruction Tuning"
_ICLR.cc/2024/Conference — ICLR 2024 spotlight_

### Official Review · Reviewer_JzMS · 2023-10-28

**Soundness:** 3 good
**Presentation:** 3 good
**Contribution:** 2 fair
**Rating:** 6
**Confidence:** 4

**Summary:**

This paper proposes MMmmoTH, which consists of two main contributions. First, it combines many different math datasets together, to the MathInstruct dataset with 260k data samples. Secondly, it uses GPT-4 to geenerate hybrid CoT and PoT solutions for the problems in the dataset. After fine tuning Llama with math instruct, their method can achieve much better results compared with the existing methods.

**Strengths:**

I think the main strength of this paper is it provides better results on many math datasets, including GSM8K and Math.

Originality: the main originality of this paper is combining multiple math dataset together, and use hybird CoT and PoT as the solution for the problems in the dataset. However, the use of CoT and PoT is a common idea used in many math papers.

Quality: Good. This paper provides a clear pipeline of the algorithm, with detailed comparison with other methods.

Clarity: Good, it is easy to follow.

Significance: Mild, as stated below.

**Weaknesses:**

This paper has limited novelty, because it seems that it mainly combines all the math datasets together, and use GPT-4 to label the dataset, and then fine tune Llama with the new labels. This is a fairly standard pipeline, and it seems that the main improvement comes from the intelligence of GPT-4.

Moreover, the idea of hybird instruction tuning is a bit confusing. According to Sec 2.4, the authors will first run PoT, and if the program cannot execute, they will switch to CoT. It seems to be a very preliminary way to combing CoT and PoT together. I was thinking a better way could be interleaving CoT and PoT in the solution.

The fine tuning part of LLama is kind of straightforward, and there are many existing work using similar ideas. So I will not say it is an important contribution.

Overall speaking, I think the main contribution of this paper is "using GPT-4 to create a new dataset (which is a combination of many existing datasets), and fine tune Llama using the created dataset". Therefore, I will say this paper has limited significance. I give weak accept mainly because I feel this is an important problem, and the authors provide a reasonably good solution.

**Questions:**

I think the authors did not provide enough details about how CoT and PoT are mixed together. It seems that both CoT and PoT are simply treated as natural languages, and feed into the model for fine tuning? Are there any special tokens used? It seems that you only used "let's write a program to solve the problem" as the prompt, right?

---

> ### Author Response · Authors · 2023-11-22
> **Author Response (1/2)**
>
> > Q: This paper has limited novelty, because it seems that it mainly combines all the math datasets together, and use GPT-4 to label the dataset, and then fine tune Llama with the new labels. This is a fairly standard pipeline, and it seems that the main improvement comes from the intelligence of GPT-4.
>
> A: Our paper focuses on showcasing the potential of open-source models like Llama to become proficient in mathematical problem-solving, demonstrating a significant leap in their capabilities without the necessity for intensive continuous training. Prior to our study, there was a notable 'math generalization ability' gap between Llama/Falcon and GPT-4, and no comprehensive empirical research had been conducted to explore the extent to which this gap could be bridged with a manageable amount of training. Our research has revealed the latent potential of Llama, illustrating that our instruction-tuned Llama model is quite adept at generalizing to unseen math datasets such as MMLU-Math, SAT, and others. This finding is crucial for the open-source community to understand the competitive edge of GPT-4 and to develop open models that can rival it.
>
> While Supervised Fine-Tuning (SFT) is a standard process, the distinctiveness of our work lies in the specific approach we took to construct the SFT data. We identified an effective recipe that mixes PoT and CoT data from diverse sources. It is an innovative method of data curation that significantly influences the performance of the model, marking a core innovation in our work.
>
> Following our study, there have been other recent significant works, such as the application of MathInstruct on Mistral-7B, which further bridge the gap. This progress demonstrates the importance and impact of our contribution, particularly for those in the open-source community striving to understand and challenge the capabilities of models like GPT-4
>
> > Q: Moreover, the idea of hybrid instruction tuning is a bit confusing. According to Sec 2.4, the authors will first run PoT, and if the program cannot execute, they will switch to CoT. It seems to be a very preliminary way to combine CoT and PoT together. I was thinking a better way could be interleaving CoT and PoT in the solution.
>
> A: There might be some confusion here. **Hybrid instruction tuning refers to the training of the LM using PoT and CoT rationale**. The hybrid decoding (first PoT, if fails, then CoT) aims to combine the accuracy of PoT’s accuracy and CoT’s coverage. We have made other attempts like “letting LM choose between CoT and PoT by itself” but the performance is not as satisfactory as “hybrid decoding”. We found that this approach despite its simplicity achieves the best accuracy in practice. Regarding “PoT + CoT interleave”, our current implementation already has a bit of this. For example, the PoT rationale actually starts with a line of comments to explain the solving process, which can be essentially seen as CoT. In multi-choice questions, we will execute the PoT to get the intermediate output and then call CoT to derive the final option. We admit that the current interleaving form is not flexible enough and we will continue to work on this to build a more cohesive decoding approach to combine CoT and PoT. Besides, we would like to point out we **tried other decoding methods like Self-consistency**. See more newly added results in the response to Reviewer z3BX.
>
> > Q: Overall speaking, I think the main contribution of this paper is "using GPT-4 to create a new dataset (which is a combination of many existing datasets), and fine-tune Llama using the created dataset". Therefore, I will say this paper has limited significance. I give weak accept mainly because I feel this is an important problem, and the authors provide a reasonably good solution.
>
> A: Thank you for your valuable feedback. In contemporary research, the composition of a dataset, often referred to as the "data recipe," is increasingly recognized as a pivotal factor in enhancing model performance. Prior to our work with MathInstruct, the understanding of how to effectively construct a dataset to augment mathematical reasoning capabilities was limited. Our unique combination of data in this study marks a significant advancement in this area. We contend that our paper sheds light on the untapped potential of open models in addressing complex mathematical problems. The insights provided in our research have already stimulated further exploration within the academic community, aiming to develop more robust Language Learning Models (LLMs). Thus, we believe the contributions of our paper extend beyond its immediate findings, fostering a deeper understanding and exploration of advanced LLMs.

---

> > ### Author Response · Authors · 2023-11-22
> > **Author Response (2/2)**
> >
> > > Q: I think the authors did not provide enough details about how CoT and PoT are mixed together. It seems that both CoT and PoT are simply treated as natural languages, and feed into the model for fine tuning? Are there any special tokens used? It seems that you only used "let's write a program to solve the problem" as the prompt, right?
> >
> > A: The training is relatively straightforward, where CoT and PoT are both treated as a natural language for next-token prediction. We made a few attempts and found that having a trigger prefix like "let's write a program to solve the problem" leads to better controllability. We tested different variants for the CoT and PoT prefix, and it turns out that different variants have little impact on the final performance.

---

### Official Review · Reviewer_x7c7 · 2023-10-31

**Soundness:** 3 good
**Presentation:** 4 excellent
**Contribution:** 3 good
**Rating:** 6
**Confidence:** 3

**Summary:**

This paper presents MAmmoTH, a set of publicly available large language models specifically created for solving a wide range of mathematical problems. These models undergo training using a meticulously assembled dataset called MathInstruct, which is compiled from 13 different math datasets and includes detailed intermediate reasoning steps. MathInstruct is unique in that it incorporates a combination of two reasoning methods, chain-of-thought (CoT) and program-of-thought (PoT), spanning diverse mathematical domains. The use of both CoT and PoT enables these models to employ distinct problem-solving approaches for different types of math challenges. This study underscores the importance of embracing diverse problem types and employing hybrid reasoning techniques to enhance the development of highly capable mathematical generalist models.

**Strengths:**

This paper provides a very simple method to synthesize useful annotations to equip small LLMs with maths reasoning ability. The authors also conduct fairly comprehensive comparison against many existing maths models with a wide range of model sizes, regardless of in-domain/out-of-domain tasks. The ablation studies help us to better understand the influence of the subparts in the training annotations.

**Weaknesses:**

Despite the commendable performance exhibited by MAmmoTH, several notable weaknesses should be acknowledged:

**Limited Technical Novelty**: The approach employed in developing MAmmoTH bears resemblance to previous works, such as Orca (Mukherjee et al., 2023). In situations where an adequate reservoir of mathematical Chain-of-Thought (CoT) or Program-of-Thought (PoT) data is unavailable, the model resorts to generating content from scratch. While this approach serves as a valuable resource and addresses the scarcity of specialized mathematics models, it may not be considered a groundbreaking contribution without a more rigorous evaluation and innovative techniques.

**Absence of an In-Depth Analysis of Training Data Distribution**: It is evident that each dataset contributes unevenly to the final training annotations. For instance, TheoremQA contains a mere 600 samples, raising questions about the true impact of incorporating such a small dataset. While it is reasonable to assume an enhancement in TheoremQA performance, it remains unclear whether this addition adversely affects the performance of other in-domain and out-of-domain tasks. Additionally, there is a concern that the model's focus on other mathematics datasets may hinder its proficiency in learning TheoremQA.

**Insufficient Ablation Studies on Training Dataset Influence**: The majority of ablation studies appear to be centered around GSM8K, with the authors incrementally augmenting the training data size atop GSM8K. However, there is a notable absence of alternative strategies, such as initially training the model on MATH and subsequently introducing other datasets in a stepwise manner. Another informative ablation study could involve the removal of individual training datasets, such as M + C + A + N, and G + M + C + N, to elucidate the specific impact of each dataset on model performance.

**Lack of Comprehensive Error Analysis**: While MAmmoTH has demonstrated impressive performance, a detailed error analysis would provide valuable insights into areas where the model can further improve. The inclusion of illustrative examples could enhance our understanding of the model's strengths and limitations, aiding in the refinement of its capabilities.

**Questions:**

1. How much of the training data would be filtered after the validation? It would be good to know the data utilization rate for the data generation process.

2. I'm curious about whether the generated annotation converted from TheoremQA can be successfully executed, because the problems there would require many advanced calculation like integral and derivative computation.

---

> ### Author Response · Authors · 2023-11-22
> **Author Response (Part 1)**
>
> > W1: Limited Technical Novelty.
>
> A: Thank you for your question on the novelty of our work. First, we would like to highlight the fact that we did compare MAmmoTH directly with Orca-Platypus (a stronger variant of the original Orca)  in Table 3 and 4. As can be seen from the results, MAmmoTH significantly outperform it on both in-domain and OOD settings. This comparison is essential in establishing the superiority of MAmmoTH. The results clearly show that MAmmoTH outperforms Orca in handling a variety of mathematical reasoning problems, underscoring the advancements our dataset MathInstruct and model MAmmoTH bring to the field.
>
> Besides, we would like to highlight our contributions and novelty again:
> - **Hybrid Rationale Integration:** A key innovation in MAmmoTH is the integration of both Chain-of-Thought (CoT) and Program-of-Thought (PoT) rationales, a feature not prevalent in models like Orca. This integration allows for flexible problem-solving strategies, enhancing computational precision and abstract reasoning capabilities in a single model​​.
> - **Extensive and Diverse Dataset Curation:** MAmmoTH leverages the MathInstruct dataset, a comprehensive and diverse compilation of mathematical problems and rationales. This dataset covers a broader range of complexity levels and mathematical fields than typically seen, addressing a critical gap in existing math datasets and contributing to the uniqueness of our work​​.
> - **Superior Performance on Comprehensive Benchmarks:** Our evaluations, as shown in Tables 3 and 4, demonstrate that MAmmoTH significantly outperforms advanced baselines like Orca-Platypus on both in-domain and out-of-domain datasets. This highlights MAmmoTH’s enhanced capabilities and generalization across various mathematical challenges​​.
> - **Detailed Ablation Studies:** The ablation studies we conducted offer insights into the impact of different training subsets and model architectures, providing a deeper understanding of how various aspects of training data influence model performance. This level of analysis adds to the scientific rigor and novelty of our work​​.
> We hope this response adequately addresses your concerns regarding the technical novelty of our work.

---

> ### Author Response · Authors · 2023-11-22
> **Author Response (Part 2)**
>
> > W2&3: Absence of an In-Depth Analysis of Training Data Distribution and Insufficient Ablation Studies on Training Dataset Influence.
>
> A: We appreciate your constructive feedback and agree that a deeper analysis of training data distribution and influence is essential for a comprehensive understanding of our model’s performance. To respond, we added additional 9 model variants trained on specific subsets and combinations in Table 5:
>
> |    Datasets                                        | GSM  | MATH| AQUA| NumG| SVA | Mat  	| Sim  	| SAT  	| MMLU| avg  	|
> |------------------------------------------------|---------|--------|----------|---------|--------|----------|---------|--------|----------|---------|
> | GSM                                            	| 56.6 	| 9.2  	| 24.4 	| 32.1 	| 65.4 	| 20.5 	| 12.3 	| 27.2 	| 25.2 	| 30.3 	|
> | GSM + MATH                                   | 58.1 	| 28.2 	| 26.0 	| 34.7 	| 64.8 	| 50.1 	| 17.1 	| 28.6 	| 28.4 	| 37.3 	|
> | GSM + MATH + TheoremQA           | 56.5 	| 26.5 	| 27.4 	| 35.5 	| 64.4 	| 50.6 	| 18.8 	| 29.1 	| 29.1 	| 37.5 	|
> | GSM + MATH + Camel                    | 57.4 	| 28.5 	| 26.2 	| 37.5 	| 65.3 	| 50.4 	| 17.7 	| 29.3 	| 28.7 	| 37.9 	|
> | GSM +   MATH + AQuA                             	| 56.1 	| 27.1 	| 37.8 	| 37.2 	| 64.8 	| 48.2 	| 19.8 	| 35.4 	| 39.8 	| 40.7 	|   	|
> | GSM + MATH + Camel + AQuA                      	| 57.5 	| 29.1 	| 46.9 	| 42.2 	| 65.8 	| 49.6 	| 32.7 	| 42.3 	| 43.1 	| 45.5 	|
> | MATH + Camel + AQuA + NumGLUE                  	| 24.7 	| 26.1 	| 39.4 	| 59.7 	| 61.6 	| 48.6 	| 43.4 	| 36.4 	| 41.2 	| 42.3 	|
> | GSM + MATH + Camel + NumGLUE                   	| 50.8 	| 26.2 	| 20.9 	| 65.5 	| 65.8 	| 48.5 	| 41.4 	| 26.4 	| 24.7 	| 41.1 	|
> | GSM + Camel + AQuA + NumGLUE                   	| 51.3 	| 14.8 	| 41.7 	| 58.8 	| 66.3 	| 31.0 	| 42.2 	| 34.1 	| 40.5 	| 42.3 	|
> | GSM + MATH + Camel + AQuA + TheoremQA          	| 55.4 	| 28.6 	| 42.5 	| 44.9 	| 65.4 	| 50.8 	| 34.9 	| 41.3 	| 42.5 	| 45.1 	|
> | GSM + MATH + Camel + AQuA + NumGLUE            	| 56.5 	| 28.9 	| 38.2 	| 63.7 	| 64.1 	| 47.9 	| 40.8 	| 38.6 	| 44.5 	| 47.0 	|
> | GSM + MATH + Camel + AQuA + NumGLUE +TheoremQA 	| 53.8 	| 27.0 	| 38.2 	| 60.8 	| 65.9 	| 50.8 	| 41.8 	| 42.5 	| 42.7 	| 47.1 	|
> | GSM + MATH + Camel + AQuA + NumGLUE + MathQA   	| 55.7 	| 28.8 	| 42.5 	| 62.1 	| 64.6 	| 45.9 	| 38.9 	| 41.3 	| 45.0 	| 47.2 	|
> | MathInstruct                                           	| 53.6 	| 31.5 	| 44.5 	| 61.2 	| 67.7 	| 46.3 	| 41.2 	| 42.7 	| 42.6 	| 47.9 	|
>
> Our key findings are summarized as follows:
> - When the model is trained on individual datasets (GSM8K,  MATH, or AQuA), it demonstrates strong performance in tasks closely related to the respective training dataset. However, it generally struggles to generalize this performance to other tasks.
> - The datasets GSM8K, MATH, and AQuA emerge as crucial components in the training process. Training the model on a combination of these datasets leads to a marked improvement in its overall performance, indicative of the complementary nature of these datasets. Omitting any one of these datasets results in a noticeable decline in overall performance.
> - TheoremQA, despite its smaller scale, exerts some influence on the model's performance. It tends to enhance the model's capabilities on datasets like Mathematics and SimulEq, yet it may slightly impede performance on GSM8K and MATH. Nevertheless, its inclusion in the training regimen is favored, as it introduces a broader variety of questions and could potentially aid the model in generalizing to other complex problems that require advanced computational skills.
> - Incorporating a wider range of datasets into the training consistently improves the model's performance across most tasks. This suggests that a broad and diverse knowledge base is instrumental in enhancing the model's ability to generalize.
>
> We have included these findings in the revision.

---

> ### Author Response · Authors · 2023-11-22
> **Author Response (Part 3)**
>
> > W4: Lack of Comprehensive Error Analysis
>
> A: Thank you for your insightful feedback regarding the need for a comprehensive error analysis of the MAmmoTH model. We acknowledge the importance of this aspect and have taken steps to enhance our evaluation by including an error analysis. We meticulously categorizes the types of 40 errors encountered by MAmmoTH on GSM8K and MATH, providing a clearer understanding of its current limitations and areas for improvement. The errors are classified into several distinct categories:
> Language Understanding Errors: These occur when the model fails to accurately comprehend the question or the context, leading to misinterpretation of the task at hand.
> Grounding Errors: Such errors arise when the model incorrectly translates quantitative data or percentages (e.g., interpreting 150% as “2 times”).
> Logic Errors: This category encompasses errors where the model's reasoning is flawed.
> Program Errors: These are errors where the model generate non-exectuable code due to the reasons like missing libraries or dependencies.
> See the image in https://imgur.com/d1imghL for breakdown.
>
> By classifying these errors, we aim to provide a clear picture of where MAmmoTH excels and where it requires further refinement. We also included specific examples of errors that MAmmoTH in the Appendix for better illustration. These examples are chosen to represent a range of issues.
>
> > Q1: How much of the training data would be filtered after the validation?
>
> A: We list the number of valid samples after filtering. AQuA-PoT (75%, 9772 out of 12881), MATH-PoT (51%, 7088 out of 14000), TheoremQA-PoT (44%, 703 out of 1600), TheoremQA-CoT (37%, 592 out of 1600), GSM8K (81%, 14591 out of 18000). We have included these statistics in the Appendix Table 6.
>
>
> > Q2: Whether the generated annotation converted from TheoremQA can be successfully executed?
>
> A: As we answered in the last question, around 44% of TheoremQA questions were successfully executed and matched the ground truth answer.

---

> > ### Comment · Reviewer_x7c7 · 2023-11-23
> >
> > I appreciate your thorough response! I'm impressed with the detailed analysis of how each data source impacts the training annotations. However, I still have reservations about the novelty of your approach. Combining the existing well-known data formulations CoT and PoT may not be considered innovative enough within the research community. While MathInstruct spans various difficulty levels in math problems, the underlying methods for obtaining CoT and PoT annotations might not be exciting. Additionally, achieving performance improvement doesn't necessarily imply a highly novel method. I may choose to keep my current score.

---

### Official Review · Reviewer_tS94 · 2023-11-08

**Soundness:** 4 excellent
**Presentation:** 4 excellent
**Contribution:** 3 good
**Rating:** 8
**Confidence:** 4

**Summary:**

The paper proposed a newly curated instruction-tunning dataset for tunning open-source models for general math problems. The MathInstruct dataset contains 13 datasets and intermediate steps to arrive at the solution. They explored a hybrid of CoT and PoT rationales. As a result, the open-source models tunned by MathInstruct beat the current best open-source models finetuned for math problems as well as GPT-4 on a portion of the math datasets.

**Strengths:**

1. The paper demonstrated that hybrid rationales sourced from both Chain-of-Thought reasoning steps and Program-of-Thought coding capability achieved better performance than using just CoT or just PoT for math problems.
2. Comprehensive results that compare models that are 1. differently sized and 2. differently instruction-tuned. This can be a very useful resource for anyone interested in studying the math reasoning capabilities of LLMs.
3. OOD scenarios are also considered and shown.

**Weaknesses:**

1. It is known that doing SFT on the dataset, especially with intermediate reasoning can do better. So even though MathInsutruct is already better than its competitors at generalizing, it can still remain a problem for even harder or unseen math problems.

**Questions:**

1. Is there a baseline that just finetunes all of the 13 datasets into 1 model? I think this would give a better idea of how to access MathInstruct. If not that's fine.

---

> ### Author Response · Authors · 2023-11-22
> **Author Response**
>
> > Q: Even though MathInsutruct is already better than its competitors at generalizing, it can still remain a problem for even harder or unseen math problems.
>
> A: We appreciate your comment on the limitations. As you rightly pointed out, and as we acknowledged in the Limitations section in the Appendix of our paper, despite the advancements brought by MathInstruct, the models might still encounter challenges with extremely complex or entirely unseen math problems. This is a common limitation when doing SFT. However, our study with MAmmoTH models demonstrates promising results in terms of performance on out-of-distribution (OOD) testing, suggesting potential pathways for enhancing model generalizability on unseen problems for future work:
>
> **Broad Coverage and Diversity in Instruction Datasets:** When constructing instruction datasets like MathInstruct, ensuring diversity is crucial. This involves covering a broad range of mathematical problems from different fields and complexity levels. Such diversity helps in training models that are better equipped to handle a wide spectrum of mathematical challenges, enhancing their generalization capabilities.
>
> **Hybrid Solution Types:** Incorporating a mix of different types of solutions, such as a combination of program-of-thought (PoT) and chain-of-thought (CoT) rationales, is more effective. This hybrid approach provides the models with varied perspectives and methods of problem-solving, which is particularly beneficial for tackling complex and novel mathematical problems.
>
> > Q: Is there a baseline that just finetunes all of the 13 datasets into 1 model? I think this would give a better idea of how to access MathInstruct. If not that's fine.
>
> A: Before our study, there was no existing baseline model that had been fine-tuned on this specific collection of datasets. The MathInstruct dataset, as introduced in our paper, is an original contribution and was specifically compiled for this research.
>
> MathInstruct is a meticulously curated dataset, comprised of 13 math datasets, including six that were newly curated by our team. This unique assembly of datasets is a novel contribution to the field of mathematical problem-solving with LLMs. The exceptional performance of our MAmmoTH models can be largely attributed to the high quality of MathInstruct. We believe that our approach of creating diverse, comprehensive datasets and incorporating a mix of solution types offers a promising direction for future research in this field.

---

### Official Review · Reviewer_z3BX · 2023-11-10

**Soundness:** 4 excellent
**Presentation:** 4 excellent
**Contribution:** 3 good
**Rating:** 8
**Confidence:** 4

**Summary:**

This paper describes MathInstruct, a mathematical reasoning-focused instruction tuning dataset assembled from 13 constituent datasets, 6 of which the authors supplement with curated synthetic rationales from GPT4. The authors use MathInstruct to train MAmmoTH, a series of Llama-based fine-tuned language models for general math reasoning. The MathInstruct dataset contains a balance of Chain-of-Thought and Program-of-Thought rationales, allowing MAmmoTH models to predict intermediate outputs in either format. The authors exploit this ability by defaulting to Program-of-Thought prediction, then backing off to CoT prompting in the event a predicted PoT program fails to execute.

The authors conduct a comprehensive evaluation of a number of open-source language models on multiple datasets, both in-domain (held out splits of datasets appearing in their MathInstruct corpus) and out-of-domain. The proposed MAmmoTH models outperform the vast majority of open LLMs, and even manage to outperform most closed LLMs (aside from GPT4) on the MATH and AQuA datasets.

**Strengths:**

- Including both CoT and PoT intermediate reasoning in MathInstruct is a great move, as the strengths and weaknesses of the two techniques are largely complementary; the hybrid inference strategy exploiting both modes is both intuitive and effective, which is very satisfying.
- Focusing on the diversity of the corpus seems to have paid off, as Table 5 shows that MathInstruct fine-tuning outperforms the sum of its largest parts on the more challenging out-of-domain test sets like the SAT questions.
- It's always nice to see performance gaps between proprietary and open models closing.

**Weaknesses:**

- It would be useful to have numbers for some of the closed models (that have APIs) for NumGlue, Mathematics, SimulEq and MMLU-Math. I realize this is a slightly annoying request as it incurs monetary costs and the scientific value of comparing to systems with unknown training distributions and architectures is dubious, but knowing which of these datasets are the most challenging for widely-used "flagship" models still has value as a heuristic for contextualizing the contribution.
- The authors only experiment with one approach to hybrid prompting - it seems like a number of approaches could be viable, e.g. self-consistency across samples from both CoT+PoT, or letting the model pick which mode to operate in. If other approaches were tried but weren't effective, it would be good to see results (or at least a remark) indicating what was tried and justifying the chosen approach as the best one empirically.

**Questions:**

1) You mention that PoT is "activated" by using a particular prompt trigger. Does this mean the model won't generate PoT rationales without specifying this trigger, or will it generate a mix of both by default? (Insight on this question could address part of my second bullet in the Weaknesses section)
2) Connected to the question above, it would be nice to add an overall ratio of PoT/CoT to Table 1. Summing and dividing the listed sizes gives 27% PoT, which explain the need for the PoT trigger prompt, but I couldn't find this number in the paper - did I miss it somewhere?

---

> ### Author Response · Authors · 2023-11-22
> **Author Response (1/2)**
>
> > W1: It would be useful to have numbers for some of the closed models (that have APIs) for NumGlue, Mathematics, SimulEq and MMLU-Math. I realize this is a slightly annoying request as it incurs monetary costs and the scientific value of comparing to systems with unknown training distributions and architectures is dubious, but knowing which of these datasets are the most challenging for widely-used "flagship" models still has value as a heuristic for contextualizing the contribution.
>
> A: We have now added GPT-4's performance data on NumGlue, Mathematics, SimulEq, and MMLU-Math, enriching our results for a more comprehensive analysis. This additional data not only sheds light on the difficulty level of each dataset but also positions our open-source models’ performance (including ours) in relation to the well-regarded, proprietary model. Notably, while there remains a performance disparity between open-source models and GPT-4, it's encouraging to note that both MAmmoTH and MAmmoTH-Coder exhibit competitive, and in some cases superior, results on specific datasets (e.g., scoring 43.6 vs. 42.5 on MATH, 65.4 vs 60.8 on DeepMind Mathematics (Mat), and 74.4 vs. 74.7 on NumGLUE). We are hopeful that our investment in presenting GPT-4's results, though financially substantial, will significantly contribute to the research community's understanding of the progress in proprietary models for mathematical reasoning tasks and offer a well-rounded perspective on our contributions.
>
> | Model                                | GSM  | MATH | AQuA | NumG | SVA  | Mat  | Sim  | SAT  | MMLU | AVG  |
> |--------------------------------------|------|------|------|------|------|------|------|------|------|------|
> | GPT-4                                | 92.0   | 42.5 | 72.6 | 74.7 | 97.0   | 60.8 | 83.1 | 95   | 74.9    | 77.0    |
> | Open-sourced SOTA on each dataset    | 81.6 | 22.7 | 51.2 | 55.4 | 71.8 | 34.5 | 37.9 | 55.9 | 52.5 | 51.5 |
> | MAmmoTH-Coder-34B                    | 72.7 | 43.6 | 54.7 | 71.6 | 84.3 | 65.4 | 51.8 | 60.9 | 53.8 | 62.1 |
> | MAmmoTH-70B                          | 76.9 | 41.8 | 65.0 | 74.4 | 82.4 | 55.6 | 51.4 | 66.4 | 56.7 | 63.4 |
>
>
> > W2: The authors only experiment with one approach to hybrid prompting - it seems like a number of approaches could be viable, e.g. self-consistency across samples from both CoT+PoT, or letting the model pick which mode to operate in. If other approaches were tried but weren't effective, it would be good to see results (or at least a remark) indicating what was tried and justifying the chosen approach as the best one empirically.
>
> A: Thanks for bringing this up. We indeed have experimented with different prompting methods. For example, we tried “letting the model decide which prompting method (PoT vs. CoT) to use” and found that the performance was not quite satisfying. In contrast, our hybrid decoding (“first PoT, if fail, then CoT”) despite its simplicity leads to better performance in practice. We are happy to put more of our exploration results to justify our choice.
>
> Conceptually, our “hybrid decoding” is a special case of self-consistency decoding, where the ensemble size equals 2 (1 PoT + 1 CoT). We further tested other more expensive self-consistency variants like 10 PoT, 5 PoT + 5 CoT, 20 PoT, and 10 PoT + 10 CoT to see how far we could reach. We lay out some additional self-consistency experimental results on GSM and MATH for MAmmoTH-Coder-7B in the Appendix :
>
> | 7B Coder Method                     | GSM  | MATH  |
> |------------------------------------------|---------|-----------|
> | 1 PoT                                        | 58.8   |   32.1   |
> | 1 PoT + 1 CoT backup              |  59.4  |   33.4   |
> | 5 PoT + 5 CoT majority vote     | 66.4   |  37.0    |
> | 10 PoT majority vote                 | 67.0   |  37.8    |
> | 10 PoT + 10 CoT majority vote | 69.6   |  38.2    |
> | 20 PoT majority vote                 | 70.1   |  38.6    |
>
> From our experiments, we show that self-consistency decoding can further improve performance. For the 7B MAmmoTH-coder, it seems that having a “PoT ensemble” already significantly improves the results to address the “non-executable issue“.

---

> > ### Author Response · Authors · 2023-11-22
> > **Author Response (2/2)**
> >
> > > Q1: You mention that PoT is "activated" by using a particular prompt trigger. Does this mean the model won't generate PoT rationales without specifying this trigger, or will it generate a mix of both by default? (Insight on this question could address part of my second bullet in the Weaknesses section)
> >
> > A: That’s a very good catch! Since we appended PoT prompt triggers (e.g., “Let’s write a program.”) in the training questions, the model would be unlikely to generate PoT solutions without such triggers. We also tried removing such triggers and let the model automatically figure out the best type of rationale for different questions but we found such a planning task is actually very difficult for the model.
> >
> > > Q2: Connected to the question above, it would be nice to add an overall ratio of PoT/CoT to Table 1. Summing and dividing the listed sizes gives 27% PoT, which explains the need for the PoT trigger prompt, but I couldn't find this number in the paper - did I miss it somewhere?
> >
> > A: Thanks! We have updated the Table 1 to reflect the PoT/CoT ratio.

---

### Official Review · Reviewer_Dja9 · 2023-11-11

**Soundness:** 3 good
**Presentation:** 2 fair
**Contribution:** 3 good
**Rating:** 8
**Confidence:** 4

**Summary:**

This paper proposed a MathInstruct dataset to improve the general performance of all types of math problems.
Specifically, they annotate public datasets with CoT and program-of-thought (PoT) annotations with GPT-4.
Through experiments on many in-domain and out-of-domain datasets, they demonstrate the effectiveness of this dataset.

**Strengths:**

1. Expensive efforts in creating the dataset, very valuable if provided to the research community.
2. Performance improvements over the baseline approaches without the MathInstruct dataset

**Weaknesses:**

1. I actually train a 7B-CodeLLaMA with GSM8K0PoT training set prompted from GPT-3.5-turbo myself, the performance can definitely achieve 62% (>59.4 in Table 3). I'm not sure what would be the quality of this dataset. I don't mind sharing my GSM8k training set for the authors to reproduce.
2. In other words, I think the author should have more experiments to justify the datasets. For example, GSM8K PoT, we need to compare with a CodeLLaMA that trained on GSM8K PoT training set.

**Questions:**

1. GSM8K has about 7k training data, how do you get 14k examples?

---

> ### Author Response · Authors · 2023-11-22
> **Author Response**
>
> > W1&2: I actually train a 7B-CodeLLaMA with GSM8K0PoT training set prompted from GPT-3.5-turbo myself, the performance can definitely achieve 62% (>59.4 in Table 3). I'm not sure what would be the quality of this dataset. I think the author should have more experiments to justify the datasets. For example, GSM8K PoT, we need to compare with a CodeLLaMA that trained on GSM8K PoT training set.
>
> A: Thank you for your insightful observations and queries regarding the performance of CodeLlama - 7B on GSM8K. In response to your query, we have added additional experimental results in the Appendix Table 7:
>
> |                                              | GSM8K | MATH | AQUA | NumG | SVA  | Mat  | Sim  | SAT  | MMLU | avg  |
> |--------------------------------------|-------|------|------|------|------|------|------|------|------|------|
> | CodeLlama - 7B - GSM8K         | **63.2**  | 18.4 | 25.4 | 32.9 | 67.6 | 40.1 | 16.3 | 26.4 | 26.1 | **35.2** |
> | CodeLlama - 7B - MathInstruct  | 59.4  | 33.4 | 47.2 | 66.4 | 71.4 | 55.4 | 45.9 | 40.5 | 48.3 | **52.0** |
>
> From the results, it is evident that training the CodeLlama 7B on the **individual GSM8K dataset can indeed yield better in-domain performance** compared with training on MathInstruct (**63.2 vs. 59.4**). This performance aligns with your observations from your own training experiments, and we appreciate you providing that as a baseline for comparison.
>
> However, we also observe that training exclusively on a single dataset like GSM8K leads to challenges in generalizing to other datasets. The average performance of the CodeLlama - 7B model is significantly higher when trained on MathInstruct (52.0%) compared to being trained solely on GSM8K (35.2%). This demonstrates a substantial improvement in the model's generalization capability when trained on the diverse and comprehensive MathInstruct dataset. We report similar results in Table 5 based on the Llama-2 7B model.
>
> These additional results provide strong evidence for the **quality and effectiveness of the MathInstruct dataset** in enhancing the performance of models across a range of mathematical reasoning tasks. While dataset-specific training can be effective for targeted tasks, the diverse and comprehensive nature of MathInstruct makes it a valuable resource for developing models with broad generalization capabilities in the domain of mathematical problem-solving.
>
>
> > Q1: GSM8K has about 7k training data, how do you get 14k examples?
>
> A: Regarding your question about how we obtained 14k examples from GSM8K, which originally has about 7k training data: For each GSM8K question, we prompted GPT-4 to generate up to 3 PoT solutions. We further enhanced the dataset quality by meticulously removing near-duplicated solutions, ensuring a diverse range of unique problem-solving approaches. These GPT-4 synthesized programs were then rigorously vetted by comparing their executed results with human-annotated ground truth, thereby ensuring the high quality and reliability of the added rationales. We’ve added these clarifications in the revision.
>
> We are grateful for your valuable feedback! Should there be any further queries or discussions needed, we are more than willing to engage and provide additional clarifications.

---

> > ### Comment · Reviewer_Dja9 · 2023-11-23
> >
> > Thanks for the response. I have adjusted my score, and thanks for the efforts to build this dataset for the research community.
> >
> > Some follow-up questions,
> >
> > 1. If I train CodeLlama - 7B on __MATH__, is it going to be better than 33.4?
> > 2. If training on the MathInstruct dataset will be harmful compared with training solely on a specific dataset, that seems not so robust in the Math domain (though only happens in __GSM8K__) for now. I hope the authors can further refine to gain improvements on all datasets.

---

> > > ### Author Response · Authors · 2023-11-23
> > > **Further Dicussion**
> > >
> > > Hi there, thanks for raising the score!
> > >
> > > 1. We haven't exactly tried CodeLlma only-MATH vs. MathInstruct. But we have tried Llama only-MATH vs. MathInstruct. The results are in table 5. It indicates that MathInstruct significantly outperforms only-MATH by 31.5% vs. 25.5%. These benefits also happen to other datasets. To the best of our knowledge, gsm seems to be the only case where the model is trading off performance.
> > > 2. Thanks for the suggestion. We will further investigate into this to better understand the reason.

---

> > > > ### Comment · Reviewer_Dja9 · 2023-11-23
> > > >
> > > > Thanks. Look forward to that.
> > > >
> > > > Also, if possible, I think we/readers would definitely like to see what would be the cost (in USD) to build this dataset if you can indicate that in the paper. I believe it is really challenging for small organizations or research labs with insufficient funding to build this.

---

### Author Response · Authors · 2023-11-22
**ChangeLog**

We would like to thank the reviewers for providing the constructive feedback. We revise our paper to reflect some of the suggestions made by the reviewers:
1. We enrich the dataset collection section to detail more about our collection strategy.
2. We add GPT-4 results for all the eval datasets for better comparison (in Table 3 and Table 4).
3. We add more ablation studies to show the influence of different subset of MathInstruct (in Table 5).
4. We add more key insights of our experiments in "Appendix C"
5. We add additional experimental results to address reviewers' questions in "Appendix D"

---

### Meta-Review · Area_Chair_bHmo · 2023-12-02

**Metareview:**

This paper presents a dataset for tuning math problem-solving models as well as a new set of open-source LLMs for math, MAmmoTH, trained on this dataset.  The models are optimized both for chain-of-thought (NL) and program-of-thought (code) reasoning.  Beyond aggregating existing datasets in MathInstruct, the paper also presents new rationales derived from GPT-4.  On "in-domain" datasets, MAmmoTH outperforms all open models of equivalent size by a significant margin, and the largest model is only 5% worse than GPT-4 on average (most of that coming from GSM8k and AQuA). Similarly strong gains over open models are shown on out-of-domain datasets.

The reviewers agree that this dataset is a valuable investment, and also praised the CoT/PoT hybrid strategy. The experiments in the paper are very thorough and it is clearly written. More error analysis and ablations were added in the response period.

The main critiques of the paper are that similar ideas have been explored before in other papers like Orca, and the general idea of tuning on a bunch of GPT-4-derived data has been used before.  In my view, the contribution of this artifact is significant even if the general space has been looked into previously. The authors do a good job of articulating this argument in response to JzMS, and in particular arguing why showing mathematical reasoning capabilities is important.

**Justification For Why Not Higher Score:**

Although the scores on this paper are high, it does roughly follow in the vein of many other papers producing customized LLMs by instruction-tuning on certain domains. I do think it deserves a spotlight, as given the interest in mathematical reasoning, many researchers at ICLR would find it useful for their ongoing work.

**Justification For Why Not Lower Score:**

This paper is clearly above the bar as a piece of empirical work, despite not having any earthshattering ideas. I can't argue too strongly that it shouldn't just be a poster, though.

---

### Decision · Program_Chairs · 2024-01-16

Accept (spotlight)